# Stem Cell Therapy Approaches for Ischemia: Assessing Current Innovations and Future Directions

**DOI:** 10.3390/ijms26136320

**Published:** 2025-06-30

**Authors:** Changguo Ma, An Yu, Tingyan He, Yulin Qian, Min Hu

**Affiliations:** Yunnan Key Laboratory for Basic Research on Bone and Joint Diseases, Kunming University, Kunming 650214, China; machangguo@kmu.edu.cn (C.M.); anyu@kmu.edu.cn (A.Y.); tingyanhe8731@163.com (T.H.); yulinqian2025@163.com (Y.Q.)

**Keywords:** stem cell, ischemic, therapy mechanisms, clinical outcome

## Abstract

Characterized by insufficient blood supply leading to tissue hypoxia and damage, ischemia is the underlying cause of major conditions such as ischemic stroke, myocardial infarction, and peripheral artery disease. Stem cell therapy, as a regenerative strategy, demonstrates significant potential in restoring tissue blood flow and organ function in ischemic environments. This review systematically explores the latest advances in stem cell therapy for ischemic diseases, focusing on different cell types and their mechanisms of action, including direct differentiation, paracrine signaling, immunomodulation, and microenvironment regulation. Furthermore, it highlights innovations in gene editing and bioengineering technologies that enhance cell delivery, targeting, and therapeutic efficacy. Simultaneously, this article discusses the challenges faced, advances in cell tracking and delivery, and future research directions, aiming to provide insights for the development of more effective and personalized treatment strategies

## 1. Introduction

Ischemic diseases encompass a group of complex syndromes caused by local blood supply insufficiency, leading to tissue hypoxia and damage. These diseases are typically categorized into central types, such as stroke, and peripheral types, including myocardial ischemia, limb ischemia, and mesenteric ischemia. The rising prevalence of ischemic diseases presents a significant challenge to global healthcare systems.

According to the 2019 Global Burden of Disease study, stroke remains a leading cause of disability-adjusted life years (DALYs) among individuals aged 50 and above, with ischemic stroke accounting for approximately 85% of all stroke cases [1]. Peripheral arterial and aortic diseases (PAAD) also affect over 113 million people worldwide aged 40 and above, with a prevalence of about 1.5% [2]. Notably, many PAAD patients suffer from chronic limb-threatening ischemia, facing severe health risks and diminished quality of life. According to the World Health Organization, ischemic diseases account for approximately 30% of global deaths (World Health Organization Global Health Estimates).

Current treatments primarily include mechanical thrombectomy and pharmacotherapy. Mechanical thrombectomy, especially effective for large vessel occlusion in acute ischemic stroke [3], physically removes thrombi to restore blood flow. However, surgical interventions carry risks and are not suitable for all ischemic conditions [4]. Conventional drugs, such as antiplatelets and vasodilators, often provide symptomatic relief but fail to fundamentally reverse tissue ischemia [5,6,7]. Limitations related to disease progression, patient heterogeneity, and side effects result in suboptimal long-term outcomes [8]. Hence, innovative and fundamental therapeutic strategies are urgently needed.

Cell therapy, i.e., introducing cells with regenerative capabilities, offers direct repair or replacement of damaged tissues at ischemic sites. As an emerging regenerative medicine approach, cell therapy holds promise for fundamentally improving long-term prognoses in ischemic diseases [9,10]. This review aims to comprehensively evaluate current innovations in cell therapy for ischemia, exploring cell types, gene editing applications, bioengineering optimizations, and clinical translation. We focus on mechanisms by which cell therapy improves ischemic microenvironments, promotes angiogenesis, and repairs tissue, while also discussing future prospects.

## 2. Overview and Pathogenesis of Ischemic Diseases

### 2.1. Types of Ischemic Diseases

Ischemic diseases can be divided into various types based on the affected organ or tissue, with the most common and clinically significant including stroke, myocardial infarction, and peripheral artery disease. This article will focus on outlining these three common clinical types of ischemic diseases.

#### 2.1.1. Ischemic Stroke

Stroke, also referred to cerebral stroke, is a sudden neurological impairment resulting from an interruption of blood supply to the brain. It may be categorized into ischemic and hemorrhagic types based on etiology. Ischemic stroke is the most prevalent, accounting for roughly 62.4% of all cases [1]. It occurs when cerebral blood vessels become blocked, leading to brain tissue ischemia and infarction. Such blockages typically stem from thrombosis or embolism [11]. Stroke is a serious disease that can lead to long-term disability or even death; it is one of the main causes of disability among adults worldwide. According to data from the World Health Organization, about 15 million people suffer from stroke globally each year, with approximately one-third of them dying as a result and another one-third left with permanent disabilities. This places a huge economic and psychological burden on patients, their families, and society [12].

#### 2.1.2. Acute Myocardial Infarction (AMI)

Myocardial infarction, commonly known as a heart attack, is caused by the blockage of the coronary arteries that supply the heart, leading to myocardial ischemia and necrosis. The most common cause is coronary atherosclerosis, which is the accumulation of cholesterol and other substances on the arterial walls to form plaques [13]. When a plaque ruptures and a bloods clot forms, it can suddenly block the artery, causing oxygen deprivation to the heart muscle. According to statistics, about 17.9 million people die from cardiovascular diseases worldwide each year, a large proportion of which are due to myocardial infarction [14].

#### 2.1.3. Peripheral Artery Disease

Peripheral artery disease (PAD) is a condition that affects arteries outside the heart or brain, most commonly those in the legs, leading to limb ischemia. Similar to myocardial infarction, the main cause of PAD is atherosclerosis [15]. The narrowing or blockage of leg arteries reduces blood flow to the muscles, causing intermittent claudication—pain in the legs that occurs during walking. In severe cases, PAD progresses to chronic limb ischemia, characterized by rest pain, ulcers, and gangrene, which may necessitate amputation. Over 200 million people worldwide are affected by PAD [16], and its prevalence continues to rise with an aging population and increasing risk factors such as diabetes [17].

### 2.2. Pathophysiology

The common pathophysiological feature of ischemic diseases is the obstruction of arterial blood flow, resulting in damage to downstream tissues due to inadequate oxygen and nutrient supply. The main cause of blood flow obstruction is atherosclerosis [18], where substances such as lipids and inflammatory cells accumulate within the arterial wall to form plaques that narrow the arterial lumen [19]. When these plaques rupture, they activate the coagulation system to form a thrombus, which, in turn, directly blocks the artery [20]. Furthermore, emboli, such as detached thrombi, fat, or air from other parts of the body, can travel through the bloodstream to obstruct smaller distal vessels [21]. 

Once blood flow is interrupted, the ischemic tissue undergoes a series of complex pathophysiological changes. Firstly, hypoxia leads to a sharp decline in cellular energy (ATP) production, impairing the normal function of ion pumps on the cell membrane, causing an accumulation of intracellular sodium and calcium ions [22]. Excessive intracellular calcium activates various destructive enzymes, damaging cell membranes, proteins, and DNA [23]. Particularly in cerebral ischemia, neurons excessively release excitatory neurotransmitters like glutamate, further exacerbating calcium influx and neuronal damage [24]. Concurrently, mitochondrial dysfunction results in the production of large amounts of reactive oxygen species (ROS), highly reactive molecules that attack various vital cellular components. Ischemia also triggers an inflammatory response, attracting numerous inflammatory cells (such as neutrophils and macrophages) to infiltrate the ischemic area [25], and inflammatory mediators released by these cells further aggravate tissue damage [26]. Ultimately, these multiple cellular stresses induced by ischemia lead to cell death and tissue dysfunction.

## 3. Drugs for Treating Ischemia

For different types of ischemic diseases and disease stages, the drugs used clinically can be broadly classified into the following categories: thrombolytics, antiplatelet drugs, anticoagulants, vasodilators, neuroprotective agents, and other symptomatic treatment drugs (Table 1).

### 3.1. Anticoagulants

Anticoagulant drugs prevent the formation and expansion of thrombi by inhibiting platelet aggregation and blocking the coagulation cascade [27]. Common anticoagulants include heparin and its derivatives (e.g., low molecular weight heparin) and vitamin K antagonists like warfarin [28,29]. Heparin acts rapidly by activating antithrombin, inhibiting various coagulation factors, and is used for acute thrombosis [30]. Warfarin inhibits vitamin K-dependent coagulation factor synthesis, has a slower onset, and is suited for long-term prevention of thrombus recurrence [31]. Novel oral anticoagulants (NOACs), such as direct thrombin inhibitors (dabigatran) and factor Xa inhibitors (rivaroxaban, apixaban), are increasingly used due to their convenience and lack of need for routine coagulation monitoring [32,33].

### 3.2. Thrombolytics

Thrombolytic drugs, or clot-dissolvers, are crucial for treating acute ischemic diseases by rapidly dissolving formed thrombi to restore vascular blood flow. Recombinant tissue plasminogen activator (rt-PA), like alteplase, is the most common [34]. Rt-PA converts plasminogen to plasmin, which hydrolyzes fibrin to dissolve thrombi. For acute ischemic stroke, administering intravenous rt-PA within a few hours of onset (typically 4.5 h) can salvage the ischemic penumbra and reduce neurological damage [35]. However, thrombolytic therapy also carries a risk of bleeding, especially intracranial hemorrhage [36], necessitating strict adherence to indications. Research continues on improving rt-PA and developing new thrombolytics to enhance efficacy and reduce bleeding risks, thereby improving outcomes for ischemic diseases.

### 3.3. Vasodilators

Vasodilators, encompassing nitrate drugs and calcium channel blockers (CCBs), treat ischemic diseases by improving local blood flow and relieving vasospasm. Nitrates (e.g., nitroglycerin) achieve this by releasing nitric oxide, relaxing vascular smooth muscle, dilating vessels, reducing cardiac load, and improving myocardial blood supply [37,38]. Their efficacy is influenced by vascular health and may cause hypotension [39]. CCBs, also potent vasodilators [40], block calcium influx into smooth muscle cells, directly dilating coronary and peripheral vessels. Dihydropyridine CCBs (e.g., nifedipine) primarily target blood vessels for hypertension and vasospasm-induced ischemia [41]. Non-dihydropyridine CCBs (e.g., verapamil) additionally slow heart rate and reduce myocardial contractility, beneficial for arrhythmias and angina [42]. Both vasodilator classes require careful clinical use due to potential adverse effects like hypotension. Non-dihydropyridine CCBs can also cause bradycardia and are unsuitable for some heart failure patients [43]. Therefore, the specific choice of vasodilator depends on the individual patient’s condition and comorbidities.

### 3.4. Neuroprotective Agents

Neuroprotective agents are receiving increasing attention in the treatment of ischemic stroke. The aim is to protect nerve cells from the effects of ischemia-reperfusion injury, salvage the ischemic penumbra, and improve disease prognosis. Neuroprotective agents typically target multiple cellular and molecular pathways, including reducing excitotoxicity, inhibiting free radical damage, regulating inflammatory responses, and inhibiting apoptosis [44]. Current drug exploration is focusing on N-methyl-D-aspartate (NMDA) receptor antagonists, antioxidants, signaling pathway modulators, and anti-inflammatory drugs [45].

### 3.5. Antiplatelet Drugs

Antiplatelet drugs inhibit thrombosis and improve ischemic symptoms by targeting different stages of platelet activation and aggregation. Their main mechanisms are the following: inhibiting platelet activation signaling pathways; Cyclooxygenase-1 (COX-1) inhibitors, such as aspirin; irreversible acetylate COX-1; blocking the conversion of arachidonic acid to thromboxane A2 (TXA2) [46], a potent platelet activator and vasoconstrictor; and inhibiting TXA2 production to reduce platelet aggregation. P2Y12 receptor inhibitors, such as clopidogrel, prasugrel, and ticagrelor, antagonize the P2Y12 receptor on the platelet surface by different mechanisms, blocking ADP-induced platelet activation and aggregation [47]. Additionally, glycoprotein IIb/IIIa (GPIIb/IIIa) receptor antagonists, including tirofiban, eptifibatide, and abciximab, directly block the GPIIb/IIIa receptor on platelets [48], which is the final common pathway for platelet aggregation. Inhibiting this receptor strongly prevents the ultimate platelet aggregation; however, due to the high risk of bleeding [49], these agents are typically reserved for high-risk situations such as acute coronary syndrome [50].

Despite showing potential in animal experiments, the application of neuroprotective agents that have been proven effective and are widely applied in acute ischemic stroke is very limited in large-scale human clinical trials [45,51]. Edaravone, a well-studied neuroprotective agent, effectively scavenges free radicals to reduce lipid peroxidation caused by reactive oxygen species (ROS), protecting nerve and vascular endothelial cells from ischemia-reperfusion injury [52]. Clinical studies show that edaravone significantly improves functional recovery in acute ischemic stroke patients with a good safety profile [53]. Despite progress, most neuroprotective agents have limited clinical translation, likely due to onset timing, therapeutic windows, and the complexity of ischemic mechanisms. Optimizing drug design and usage remains a key focus for future research.

### 3.6. Limitations of Pharmacological Treatments

Although existing drugs play an important role in the treatment of ischemic diseases, their efficacy and applicability still have significant limitations. First, many anticoagulant and thrombolytic drugs have a high risk of causing bleeding complications (such as intracranial hemorrhage). These risks significantly limit the time window for drug use. The efficacy of vasodilators and CCBsLVEF is usually limited, mainly improving hemodynamics without directly affecting the cellular and molecular mechanisms of ischemic injury.

In addition, existing drugs often fail to effectively address the multiple pathophysiological mechanisms of ischemia. On the one hand, the complexity of ischemic diseases is reflected in the involvement of multiple interacting links such as blood flow obstruction, cellular metabolic disorders, inflammatory responses, free radical damage, and cell apoptosis. Single-target drugs usually only play a limited role in specific links and often fail to significantly improve the overall prognosis. These limitations indicate that developing new comprehensive therapies is an urgent challenge.

## 4. Progress and Mechanisms of Stem Cell Therapy

Cell therapy for ischemia is one of the research hotspots in the field of regenerative medicine and is currently in a phase of continuous development and exploration. With in-depth studies on cell types, mechanisms of action, and optimal delivery methods, cell therapy has shown great potential in promoting angiogenesis, improving tissue function, and facilitating injury repair. It has become an important approach for the treatment of ischemic diseases.

### 4.1. Types of Stem Cells Utilized in Ischemic Treatment

Mesenchymal Stem Cells (MSCs): Mesenchymal stem cells (MSCs) are adult stem cells with self-renewal and multipotent differentiation potential, mainly derived from bone marrow, umbilical cord, and adipose tissue. MSCs can differentiate into various cell types and possess immunomodulatory functions, which provide a good basis for the repair and regeneration of ischemic tissues [54]. Studies have shown that MSCs exhibit effective therapeutic effects in the treatment of myocardial infarction [55], ischemic stroke [56], and limb ischemia [57], highlighting their promising application prospects. In addition, MSCs have good safety and low immunogenicity, making them an ideal cell type for ischemic therapy.

Neural Stem Cells (NSCs): NSCs have the ability to self-renew and differentiate into multiple cell types, including neurons, astrocytes, and oligodendrocytes. They show great potential in the treatment of cerebral ischemia (stroke). NSCs play a key role in neural tissue repair through cell replacement and the secretion of bioactive factors that can modulate neuroregeneration, immune response, and the integrity of the blood–brain barrier [58]. However, challenges include low survival rates after transplantation due to the harsh ischemic microenvironment [59,60] and limited cell sources, such as the ethical and technical issues surrounding embryonic stem cells and fetal-derived neural stem cells [61]. NSCs derived from direct reprogramming avoid the use of embryonic cells and reduce the risk of immune rejection [62], but the efficiency of reprogramming and the stability of cell function still need to be optimized.

Embryonic Stem Cells (ESCs): ESCs can differentiate into various cell types, such as cardiomyocytes [63], endothelial cells [64], and smooth muscle cells [63], promoting tissue repair and functional recovery. In addition, small extracellular vesicles (sEVs) [65] and conditioned media derived from ESCs [66] also show promising therapeutic effects by modulating immune responses and improving neurological function, providing new strategies for the treatment of ischemic diseases. Due to safety concerns, current research prioritizes the paracrine mechanism and the differentiation of ESCs into specific cell types before transplantation. For example, in treating ischemic myocardial infarction, ESCs can be induced to differentiate into cardiomyocytes and endothelial cells. Additionally, ethical considerations further limit the clinical application of ESCs.

Induced Pluripotent Stem Cells (IPSCs): iPSCs are generated by reprogramming adult cells, such as skin. iPSCs possess strong differentiation capabilities and can participate directly in angiogenesis in ischemic tissues by differentiating into endothelial progenitor cells [67] and cardiomyocytes [68]. They show great potential in treating ischemic diseases while circumventing the ethical issues associated with embryonic stem cells and allowing for personalized treatments, thereby reducing immune rejection risks [69,70]. It is crucial to differentiate induced pluripotent stem cells (iPSCs) into specific cell types, such as endothelial cells. Research has revealed that the progesterone receptor (PR) is constitutively expressed in iPSCs and plays an important role in iPSC reprogramming and differentiation [71]. This suggests that regulating PR may induce iPSCs to differentiate into specific cell types. Future research investigating the regulation of PR to induce iPSC differentiation into endothelial and smooth muscle cells is warranted. However, challenges remain, including tumorigenic risks from undifferentiated iPSCs that may lead to teratomas [72,73], as well as difficulties in achieving large-scale, standardized production of iPSC-derived products that meet clinical standards [74]. Key issues to address include safety, cell survival rates, and differentiation efficiency.

Hematopoietic stem cells (HSC): HSCs are a type of multipotent stem cell found in hematopoietic tissues such as bone marrow, fetal liver, and umbilical cord blood. They possess the abilities of self-renewal and multi-lineage differentiation and can give rise to all types of mature blood cells, including red blood cells, white blood cells, and platelets. The functions of HSCs are not limited to hematopoiesis [75]; they also play roles in tissue repair and angiogenesis, showing great potential in the treatment of ischemic diseases [76]. The mechanisms by which HSCs treat ischemic diseases mainly include vascular regeneration, paracrine effects, and tissue repair and regeneration. HSCs can differentiate into endothelial cells to promote neovascularization [77,78] and secrete various growth factors to activate angiogenic responses in local tissues [79], thereby improving blood flow to ischemic areas. In addition, HSCs can regulate the local inflammatory and immune microenvironment, supporting the repair and functional recovery of damaged tissues [80], thus playing an important role in the treatment of ischemic diseases. Clinically, HSCs have been explored in conditions of myocardial ischemia, demonstrating the potential to repair damaged myocardium and improve cardiac function [81]. However, HSC-based treatment for ischemia still faces many challenges, including limited sources of HSCs, difficulties in obtaining high-quality HSCs, issues with the standardization of manufacturing processes [82], and the need for long-term safety validation.

Cardiac stem cells (CSC): CSCs are a type of multipotent cell residing in heart tissue, capable of differentiating into cardiomyocytes, smooth muscle cells, and endothelial cells after cardiac injury. Owing to their ability to specifically differentiate into cardiomyocytes [83], CSCs have demonstrated high therapeutic efficacy in the treatment of ischemic heart disease. CSCs can differentiate into cardiomyocytes, vascular endothelial cells, and smooth muscle cells, thereby replacing necrotic myocardium and improving microcirculation in the heart [84,85]. Additionally, CSCs secrete various growth factors that promote cardiomyocyte survival and angiogenesis and can inhibit immune cell activity to reduce inflammatory damage and decrease cardiomyocyte death [86]. Clinical studies have shown that endogenous cardiovascular progenitor cells directly isolated from the heart can provide cell therapy for cardiac patients, significantly improving cardiac function and patient prognosis [87].

The main challenges with CSCs are their scarcity and the difficulty in isolating them. CSCs were initially identified by the expression of c-kit (stem cell factor receptor). However, research has revealed that only a very small proportion of c-kit-positive cells in the heart possess the true phenotype and differentiation potential of pluripotent CSCs [83,88]. This makes it difficult to isolate and purify CSCs. Previous studies developed an experimental method to isolate and expand adult cardiac stem cells (in the form of cardiospheres) from human heart biopsy samples and mouse hearts. The broad applicability of this method marks a milestone in the clinical translation of autologous cell therapy [89]. Cardiosphere-derived cells (CDCs) are cells isolated from these cardiospheres, possessing self-renewal and being able to differentiate into cardiomyocytes, endothelial cells, and smooth muscle cells [90]. Clinical trials have evaluated the safety and efficacy of autologous and allogeneic CDCs in patients with ischemic heart disease. The results showed that autologous CDCs significantly reduced scar size, increased viable myocardium, and improved infarcted region function at 6 and 12 months [91], whereas the allogeneic CDCs, though not achieving similar scar size reduction, did enhance left ventricular diastolic function [92]. CDCs have received widespread attention in cardiac regenerative medicine and are considered a potential cell source for cardiac repair.

Stem cells, including Mesenchymal Stem Cells (MSCs), Neural Stem Cells (NSCs), Embryonic Stem Cells (ESCs), and Induced Pluripotent Stem Cells (iPSCs), offer significant promise for treating ischemic diseases by promoting tissue repair and functional recovery (Table 2). While all these stem cell types contribute to the advancing field of ischemic therapy, MSCs are currently the most prominent and extensively utilized stem cell type for treating a variety of ischemic diseases, owing to its pronounced paracrine effects, strong immunomodulatory capabilities, and relatively low safety risks.

### 4.2. Mechanisms of Action

Stem cells play a multi-level and complex role in the treatment of ischemic diseases. Through direct differentiation, paracrine effects, immune regulation, and interaction with the extracellular matrix, stem cells effectively promote the repair and regeneration of ischemic tissues (Figure 1). The following are the main mechanisms by which stem cells promote tissue repair:

#### 4.2.1. Direct Differentiation into Functional Cells

Stem cells play a direct role in vascular remodeling and tissue reconstruction by differentiating into the specific functional cells lost or damaged in ischemic tissues. They can directly differentiate into vascular endothelial cells and smooth muscle cells [93], actively participating in the reconstruction and stabilization of blood vessel walls, which improves vascular function and perfusion. Furthermore, stem cells are capable of differentiating into cardiomyocytes and nerve cells [94]. These differentiated cells directly contribute to the structural restoration and functional recovery of the affected tissues. A prime example is observed in myocardial infarction models, where stem cells differentiate into cardiomyocytes, vascular endothelial cells, and pericytes, thereby replacing necrotic heart muscle and damaged blood vessels, leading to improved myocardial contractility and tissue perfusion [95].

#### 4.2.2. Paracrine Effects

Stem cells release a variety of soluble factors through paracrine mechanisms, such as growth factors, cytokines, and exosomes. These factors play multiple roles in the treatment of ischemic diseases.

##### Promote Angiogenesis

Stem cells can secrete various angiogenesis-related growth factors and cytokines, such as Vascular Endothelial Growth Factor (VEGF), Fibroblast Growth Factor (FGF), Hepatocyte Growth Factor (HGF), and so on [96]. These factors promote the formation of new blood vessels and improve blood supply to ischemic tissues [97].

##### Anti-Apoptosis and Oxidative Stress

Stem cell transplantation can inhibit mitochondrial-mediated apoptosis by regulating the expression of *Bcl-2* family proteins. For example, hADSCs (human adipose-derived stem cells) that overexpress HIF-1α can suppress mitochondrial-mediated apoptosis in ischemic microenvironments, enhancing cell survival rates [98]. MSCs can also reduce apoptosis by suppressing the activation of caspase proteases through the secretion of exosomes [99]. Stem cells secrete anti-apoptotic factors (such as HGF, IGF-1), protecting cells in the ischemic environment from damage caused by oxidative stress, inflammatory factors, etc., and improving cell survival rate [100].

Oxidative stress is another key pathological factor in ischemic diseases, where excessive production of reactive oxygen species (ROS) leads to cellular damage and dysfunction. MSCs can express a variety of antioxidant enzymes, such as superoxide dismutase (SOD) and glutathione peroxidase (GPx), which directly eliminate ROS and alleviate oxidative stress-induced injury [101]. Additionally, MSCs can release cytokines that activate endogenous antioxidant pathways, such as the Nrf2 pathway [102], thereby enhancing the antioxidant capacity of tissue cells.

##### Immunomodulation

Stem cells have immune regulation functions. Stem cells can inhibit the inflammatory response caused by ischemia-reperfusion by secreting immunosuppressive factors (such as TGF-β and IL-10) to suppress pro-inflammatory cytokines [103]. In addition, stem cells can regulate the polarization state of immune cells in tissues, such as inducing macrophages to transform from pro-inflammatory M1 type to anti-inflammatory M2 type [104]. These immunomodulatory capabilities improve the ischemic microenvironment and promote tissue healing and regeneration.

##### Regulate the Microenvironment and Extracellular Matrix

Stem cells can secrete matrix metalloproteinases (MMPs) in ischemic tissues to remodel the extracellular matrix (ECM) of the ischemic tissue, providing a favorable structural environment for the adhesion, migration, and proliferation of local cells [105]. In addition, the matrix-forming factors secreted by stem cells can further induce the activation and migration of local progenitor cells or stem cells [106,107], forming a more complete repair network.

#### 4.2.3. Promote Endogenous Regeneration Mechanisms

Stem cells can activate and recruit endogenous progenitor cells or stem cells from host to the ischemic site by releasing chemokines such as SDF-1 [108]. These activated endogenous cells can participate in angiogenesis and tissue repair, thereby increasing repair efficiency [109]. For example, it has been found that transplanted bone marrow MSCs can significantly enhance the migration of host endothelial progenitor cells to the ischemic region and angiogenesis through paracrine effects [110].

The above paracrine factors work together to provide a favorable microenvironment for angiogenesis and functional recovery of ischemic tissues, which is the most important mechanism of stem cell therapy in treating ischemic diseases.

## 5. Clinical Trials and Outcomes

Stem cell therapy has clinical applications in multiple ischemic fields, such as myocardial infarction (MI), ischemic stroke, and chronic limb ischemia (CLI). Clinical results in each field may vary and need to be discussed separately (Table 3).

### 5.1. Myocardial Infarction

Myocardial infarction remains a leading cause of mortality worldwide. While current therapies have improved patient outcomes, regenerating damaged myocardium and restoring function remain significant challenges. Stem cell therapies, primarily utilizing bone marrow-derived MSCs, bone marrow mononuclear cells, HSCs, CSCs, and umbilical cord-derived MSCs (UC-MSC), are being investigated clinically for MI treatment. Multiple clinical trials and meta-analyses suggest that stem cell therapy can significantly improve cardiac function in MI patients, increasing left ventricular ejection fraction (LVEF) by approximately 4.58% and reducing left ventricular end-systolic volume (LVESV) by approximately 5.18 mL, with a dose-dependent effect [111]. Four-year follow-up data indicate that MSC treatment does not increase mortality or rehospitalization risk, suggesting acceptable long-term safety [112]. Mesenchymal stem cells (MSCs) may be more effective than bone marrow mononuclear cells (BM-MNCs) in improving LVEF, particularly in patients receiving stem cell transplantation early (less than 11 days) after acute myocardial infarction (AMI). Injection of HSCs via the coronary artery or into the infarcted myocardium can improve cardiac function, including left ventricular remodeling, perfusion after acute myocardial infarction, and New York Heart Association functional class. Furthermore, patients receiving bone marrow-derived hematopoietic stem cell therapy experienced a significant reduction in the combined endpoint of death, rehospitalization for heart failure, and re-infarction [113]. CSCs are also important cells for treating myocardial infarction (MI). Clinical results show that autologous CSCs are effective for ischemic cardiomyopathy. CSC therapy can significantly improve left ventricular function, reduce infarct size, and increase viable tissue, with effects lasting at least one year, indicating their potential for cardiac regeneration [87]. More clinical studies on CSCs are still ongoing [114,115,116]. However, some studies have not observed significant clinical benefits. A randomized, double-blind, placebo-controlled phase II clinical trial evaluating autologous bone marrow-derived MSCs for AMI showed no significant difference between the stem cell group and the placebo group in improving infarct size or cardiac function [117], potentially due to cell type, timing of transplantation, or patient heterogeneity. Future large-scale, long-term follow-up studies are needed to confirm efficacy.

The main routes of stem cell therapy for myocardial ischemia include intracoronary injection, intramyocardial injection, and intravenous injection [118]. Intracoronary injection is the most commonly used route of administration, which can directly deliver stem cells to the infarcted area. Intramyocardial injection can more accurately locate stem cells to the damaged myocardium, but the operation is more difficult [119]. Intravenous injection has the advantages of simple operation, but the retention rate of stem cells in the target organ is low [120].

### 5.2. Stroke

Stroke is one of the leading causes of disability and death worldwide. Traditional treatments have mainly focused on restoring blood flow and reducing secondary injury, but their effectiveness in long-term neurological recovery is limited. In recent years, clinical research on stem cell therapy for stroke has made significant progress, yet it also faces many challenges. On one hand, some clinical trials have demonstrated the potential of stem cell therapy. For example, a study involving autologous bone marrow-derived mesenchymal stem cell transplantation in patients with acute ischemic stroke showed that the therapy was safe and that some patients experienced functional improvements [121]. However, other studies have failed to reach positive conclusions. A multicenter, randomized, double-blind, placebo-controlled clinical trial administered allogeneic stem cells intravenously within 18 to 36 h after stroke onset; while the treatment was found to be safe, there were no significant differences between the treatment and placebo groups in primary or secondary endpoints at 90 days [122]. These differences may be related to factors such as stem cell type, dose, route of administration, patient selection, and stroke type.

The main routes of stem cell therapy for stroke include intravenous injection, arterial injection, and intracerebral injection. Intravenous injection is the most commonly used route of administration, with the advantages of simple operation, but the retention rate of stem cells in the target organ is low [123]. Arterial injection can more directly deliver stem cells to the damaged brain tissue but may increase the risk of vascular embolism [124,125]. Intracerebral injection can more accurately locate stem cells to the damaged brain tissue, but the operation is more difficult and may cause local inflammatory reactions [126].

Future research needs to be strengthened in the following areas: first, conducting larger-scale, multicenter, randomized controlled clinical trials to enhance the reliability and generalizability of study results; second, further exploring the optimal protocols for stem cell therapy, including the selection of cell types, dosage optimization, and determination of the timing and routes of administration.

### 5.3. Chronic Limb-Threatening Ischemia

Limb ischemia, especially critical limb ischemia (CLI) caused by peripheral artery disease (PAD), is a severe vascular condition often leading to pain, ulcers, and even amputation. Stem cell therapy, as an emerging treatment strategy, offers new hope for “no-option” patients. Cell therapies primarily based on bone marrow-derived mesenchymal stem cells (BM-MSCs) and adipose-derived mesenchymal stem cells (AD-MSCs) have entered early-stage clinical trials [127]. Some clinical trials have shown that stem cell treatment can improve blood perfusion in patients with limb ischemia, relieve pain, promote ulcer healing, and reduce amputation rates. For example, a randomized, double-blind study evaluating autologous bone marrow mesenchymal cells in patients with severe limb ischemia who had no surgical options demonstrated that patients receiving autologous bone marrow cell therapy had significantly lower amputation rates. Additionally, ankle-brachial index (ABI), transcutaneous oxygen pressure, resting pain, intermittent claudication pain scores, wound healing, and walking distance were all significantly improved [128]. A phase 1 clinical trial assessing the safety of human placental mesenchymal stem cells (P-MSCs) for treating CLI patients reported that intramuscular injection of placental MSCs significantly improved ABI, promoted ulcer healing, and reduced the risk of major amputation. The highest dose tested (6 × 10^7^ cells) was well tolerated, and a 6-month follow-up showed an 80% symptom relief rate [129]. These findings indicate that MSC therapy for limb ischemia is both safe and effective.

The routes of administration of stem cell therapy for limb ischemia mainly include arterial injection, intravenous injection, and intramuscular injection [128,130]. Arterial injection can deliver stem cells more directly to the ischemic site but may increase the risk of vascular embolism. Intramuscular injection is easy to operate, but the retention rate of stem cells in the target organ is low. Future research directions may include the development of a new generation of stem cell products, optimization of therapeutic protocols, and further validation of their efficacy and safety through larger-scale, multicenter, randomized controlled trials.

Clinical research on stem cell therapy for ischemia involves multiple key parameters, including cell type, dosage, time window, delivery route, and safety/efficacy. Among the various cell types, MSCs are the most widely studied due to their easy accessibility, low immunogenicity, and multi-differentiation potential. While iPSCs and NSCs show potential for nerve regeneration in stroke models, their clinical translation is in its early stage [131].

Optimizing the dosage is critical, as studies show a wide range of infusion doses from 1 × 10^6^ to 2 × 10^8^ cells for myocardial infarction [132], with stroke trials commonly using 1 × 10^6^ to 5 × 10^7^ cells [133]. The dose–response relationship is non-linear, with excessive doses potentially elevating embolism risk. The timing of treatment also plays a vital role: administering cells within 24–72 h post-myocardial infarction can help salvage ischemic myocardium [132], while for a stroke animal model, administration within two weeks can improve neurological outcomes [134]. Various delivery routes are utilized, including intravenous injection, intra-arterial injection, and local injections, each with distinct advantages and risks. Assessing the short- and long-term safety of these therapies is essential; most trials report only mild adverse effects, yet concerns such as immune rejection and abnormal differentiation remain pertinent. Ethical and regulatory challenges, including the standardization of cell sources and manufacturing processes, are crucial for advancing these therapies into clinical practice.

Future research needs to further validate the efficacy and safety of stem cell therapy through larger-scale, multicenter randomized controlled trials and optimize treatment protocols to address challenges such as cell delivery, microenvironment regulation, and standardization of therapy.

## 6. Future Directions in Cell Therapy

### 6.1. Genetic Engineering

Currently, stem cell therapy for ischemic diseases still faces multiple challenges. First, stem cells often exhibit low survival and homing rates after transplantation, mainly due to severe oxygen and nutrient deprivation and inflammatory responses in the ischemic/inflammatory microenvironment, which cause extensive cell death and limit their effective delivery to target areas. Second, the therapeutic functions of the stem cells themselves may also be limited, as their paracrine effects and differentiation capacity might be insufficient to fully reverse severe ischemic damage. Lastly, allogeneic stem cell transplantation carries the risk of immune rejection. In cell therapy for ischemic diseases, gene-editing technologies, especially the CRISPR-Cas9 system, have shown great potential in recent years in the field of stem cell therapy for ischemic diseases.

Enhance cell survival rate: The harsh microenvironment of ischemic tissues (hypoxia, inflammation, etc.) is the main reason for the death of transplanted cells. Gene editing technology can enhance the survival of stem cells in ischemic environments and promote their differentiation into specific cell types, thereby repairing damaged tissue more effectively [135]. For example, using CRISPR technology to knock out the *PTEN* gene in MSCs can activate the PI3K/Akt signaling pathway, enhance the anti-apoptotic ability of cells, and improve their survival rate in ischemic myocardium [136]. Gene editing has also demonstrated significant potential in cardiomyocyte regeneration. Genetically editing cardiac stem cells to overexpress specific growth factors or non-coding RNAs can enhance their regenerative potential in damaged myocardium [137]. For example, genetically modified CSCs secreting IGF-1 and HGF can stimulate the proliferation of endogenous eCSCs and reduce cardiomyocyte apoptosis through paracrine effects [138]. Akt gene modification can phosphorylate apoptotic proteins such as BAD and Caspase-9, inhibiting programmed cell death. As a downstream factor of Akt, overexpression of Pim-1 kinase can enhance the anti-apoptotic capacity of CSCs [139]. Overexpression of HAX1 in CSCs significantly enhances their proliferation capacity and strengthens their resistance to hypoxia-induced cell death [140].

Improve targeting and homing ability: In order to allow transplanted cells to more effectively reach ischemic tissues, the expression of chemokine receptors on the cell surface can be enhanced through gene editing technology [141], such as *CXCR4*, *CCR2*, etc. CXCR4 overexpression can enhance the chemotactic response of MSCs to CXCL12 and significantly improve their homing ability to damaged tissues [142]. In adipose-derived stem cells (ADSCs), gene editing overexpression of CCR2 significantly improved their homing ability to muscle tissues and improved the pathological repair effect [143]. These receptors can bind to chemokines released by ischemic tissues and guide cells to migrate to the lesion. In addition, cells can also be “disguised” through gene editing technology to express proteins that are highly expressed in the receptor tissue, thereby increasing the adhesion of cells. For example, expressing target tissue-specific receptors (such as tumor-highly expressed CXCR4 or CCR2) on the cell surface through genetic engineering can enhance its interaction with the lesion microenvironment [144].

Enhance angiogenesis ability: Angiogenesis is a key link in repairing ischemic tissues. Through gene editing technology, the expression of angiogenic factors such as vascular endothelial growth factor (VEGF) and hepatocyte growth factor (HGF) in transplanted stem cells can be increased, promoting the formation of new blood vessels and improving tissue blood supply [145].

Enhancing Stem Cell Paracrine Effects and Endogenous Activation: Genetically modified stem cells can secrete factors such as IGF-1, HGF, and VEGF, which activate endogenous stem cells via paracrine signaling. For example, allogeneic pig CSCs modified to secrete HGF can activate endogenous c-Kit+ CSCs in pig models, promoting autologous myocardial regeneration [146]. Akt-modified CSCs secrete SDF-1, recruiting endogenous stem cells to the injury site [139].

The combination of gene editing technology and stem cells shows great potential in treating ischemia, but it also faces many challenges. First, off-target effects, where the CRISPR-Cas9 system cuts DNA at non-target sites, can lead to potential side effects [147]. Therefore, it is necessary to reduce the risk of off-target effects by optimizing gRNA design and using high-fidelity Cas9 protein. Second, the CRISPR-Cas9 system may trigger immune responses, resulting in the clearance of transplanted cells. Finally, the application of gene editing technology also involves ethical issues.

### 6.2. Innovations in Bioengineering

In recent years, the development of bioengineering technology has provided a crucial boost for the optimization and improvement of cell therapy. By using biomaterials, exosome modification, and delivery platforms, cell delivery efficiency and functional efficacy in vivo can be significantly enhanced. This study will now specifically explore several innovative strategies and their research directions.

#### 6.2.1. Application of Biomaterials in Cell Delivery Systems

The application of biomaterials can significantly increase the survival and retention rates of stem cells in ischemic areas. For example, three-dimensional (3D) biological scaffolds, by mimicking the physical and chemical properties of the natural extracellular matrix (ECM), provide a stable microenvironment for stem cells, thereby promoting cell adhesion, proliferation, and differentiation [148]. Scaffolds based on natural biomaterials (such as collagen or hyaluronic acid) have been proven to enhance the survival of stem cells in ischemic tissues and activate local repair processes [149,150]. In addition, the use of 3D printing technology can customize scaffolds according to the individual needs of patients, optimize scaffold design, and further improve the efficiency of cell delivery [151,152].

#### 6.2.2. Hydrogel Carriers Improve Cell Therapy

Biomaterials have a significant impact on the differentiation and paracrine activity of stem cells, thereby enhancing the therapeutic efficacy of stem cells. As an important type of biomaterial, hydrogels have been widely used in cell therapy in recent years. They are characterized by high water solubility, high biocompatibility, and injectability, making them particularly suitable for minimally invasive treatments. Hydrogels can not only directly encapsulate stem cells to improve survival rate but also load angiogenic factors (such as VEGF) or small molecule drugs to further promote the repair of ischemic tissues [153,154].

#### 6.2.3. Smart Biomaterial Design

Smart biomaterials can respond to specific physiological environments, such as oxygen concentration, pH value, enzyme activity, etc., and dynamically regulate the delivery of cells or drugs. For example, hypoxia-responsive hydrogels release higher concentrations of stem cells and supporting factors in ischemic areas, thereby enhancing the therapeutic effect [155,156]. This type of material that can actively participate in tissue repair regulation has opened up a new field of biological therapy [157].

Combining cell therapy with bioengineering has broad prospects but also faces many challenges. First, it is necessary to select biomaterials with good biocompatibility and safety to avoid immune reactions and toxic reactions. Second, it is necessary to optimize the integration method of cells and biomaterials to promote cell adhesion, proliferation, and function on the materials. In addition, long-term clinical trials are needed to evaluate the long-term efficacy and safety of this combined therapy. At the same time, developing technologies for large-scale production of cells and biomaterials is also key, which will help reduce treatment costs and improve accessibility. Despite these challenges, with the continuous advancement of bioengineering technology, the combination of cell therapy and bioengineering is expected to bring revolutionary breakthroughs in the treatment of various diseases in the future.

### 6.3. Development of Exosome Delivery Systems

Exosomes are a type of nanoscale vesicles secreted by cells that can carry bioactive molecules such as proteins, RNA, and lipids [158]. Given that the majority of therapeutic effects of cells are mediated through paracrine factors, including extracellular vesicles secreted by cells (such as exosomes), researchers are actively exploring the use of exosomes as a new strategy for treating ischemic diseases. The engineered modification of exosomes further expands their application in cell therapy.

#### 6.3.1. Engineered Exosome Modification to Improve Function

Studies have shown that genetic or chemical modification can increase the targeting and therapeutic efficiency of exosomes. For example, connecting aptamers or antibody molecules to the surface of exosomes can achieve their directional delivery in ischemic areas [159,160]. In addition, by loading nucleic acid drugs (such as miRNA or siRNA), exosomes can also precisely regulate gene expression in tissues [161]. For example, encapsulating the super suppressor IκBα into exosomes can alleviate renal ischemia-reperfusion injury [162].

#### 6.3.2. Large-Scale Controllable Production Technology

In order to meet the clinical needs of exosome therapy products, bioreactors that can produce exosomes on a large scale are being developed, and purification methods (such as ultracentrifugation and membrane filtration technology) are being improved to ensure the purity and biological activity of exosomes [163].

### 6.4. Combined Treatment of Stem Cells and Drugs

The combined use of stem cell therapy and angiogenic drugs or immunomodulators has a synergistic effect. For example, drugs such as fasudil can synergistically improve cell survival and neurological function recovery in ischemic areas with bone marrow mesenchymal stem cells (BMSCs) [164]. Combining erythropoietin EPO with umbilical cord blood-derived mesenchymal stem cells (MSCs) can promote nerve regeneration and functional recovery more effectively than using EPO or stem cells alone [165]. This synergistic effect stems from the ability of drugs to enhance the paracrine activity of stem cells, while stem cells can secrete a variety of neurotrophic factors and anti-inflammatory factors, which can enhance the efficacy of drugs [166]. Therefore, combined treatment strategies can produce an additive effect beyond a single therapy through multi-target action [167].

### 6.5. Production Cost and Accessibility

The high cost and limited scalability of cell therapy are the main bottlenecks restricting its widespread application [168]. Its involves multiple steps, including cell collection, in vitro expansion, directed differentiation, quality testing, storage and transportation, and clinical application. Each step requires high-standard facilities and professional personnel, significantly increasing the overall cost. Production must comply with strict Good Manufacturing Practice (GMP) standards, necessitating the construction of highly specialized and ultra-clean production facilities, which entails substantial construction and maintenance costs [169]. Autologous cell therapy requires establishing a separate production batch for each patient, from cell collection, genetic modification, expansion to final product release. The entire process cannot be scaled up and relies heavily on manual operations by highly skilled professionals, resulting in extremely high labor costs. Although allogeneic cell therapy allows for a certain degree of scale-up production, it still faces costs related to donor screening, cell bank establishment, and cell storage and transportation [170]. Rigorous quality control is also a significant part of the cost. Every batch of cell products must undergo a series of stringent quality tests—including for sterility, purity, potency, identity, and safety—before being transfused back to the patient. These testing procedures are complex, time-consuming, and expensive. Moreover, due to the personalized nature of the products, establishing quality standards and testing methods is also challenging. The high cost of cell therapy imposes a serious burden on healthcare insurance systems and patients. For example, in the U.S. and European markets, the cost of a single cell therapy can often reach tens of thousands of U.S. dollars or more, with some customized products priced over one hundred thousand dollars [171], making it especially unaffordable for developing countries and low-income populations. Meanwhile, autologous cell therapy adopts a “one batch per person” production model of “scale-down” rather than “scale-up,” which cannot meet the demand of a large patient population. The production cycle is long; it usually takes several weeks from collecting patient cells to preparing the final product. For critically ill patients, this waiting period can be fatal. The entire process involves cold chain logistics, requiring the transport of patient cells from hospitals to production centers and then returning the prepared product back to the hospitals, which imposes extremely high demands on logistics and coordination [171].

To reduce costs and improve accessibility, it is necessary to develop more efficient and automated cell manufacturing equipment and processes in the future, minimizing manual intervention, lowering production costs, and reducing batch-to-batch variability. The development of allogeneic “off-the-shelf” products holds significant potential to substantially decrease the cost per treatment and enable immediate availability. Establishing more efficient and reliable cold chain logistics and information management systems is also crucial. At the same time, healthcare insurance policies, charitable funds, and multi-party collaborations contribute to the inclusion and wider adoption of certain high-value cell therapies. However, overall, cost-effectiveness and equitable accessibility remain critical barriers that must be prioritized in the clinical translation and large-scale deployment of cell therapies going forward.

Future directions of cell therapy are mainly focused on two major areas: gene editing technologies and innovations in bioengineering (Figure 2). Gene editing technology precisely modifies the cell genome, effectively improving the survival rate, targeting ability, and angiogenesis ability of cells in the treatment of ischemic diseases, thereby significantly enhancing the therapeutic effect. However, it still faces challenges such as off-target effects, delivery safety, immune responses, and ethical issues. In bioengineering, the application of biomaterials (such as three-dimensional scaffolds, hydrogels, and smart materials) optimizes cell delivery systems, improving cell survival and functional performance. Engineered exosomes are used to enhance targeting and therapeutic efficiency and promote the development of large-scale production technologies. In addition, the combined application of stem cells and drugs demonstrates synergistic effects, enhancing the repair capacity of ischemic tissues. Despite problems such as material safety, cell and material integration, long-term efficacy validation, and production costs, the combination of gene editing and bioengineering is considered a key path for achieving breakthroughs in the field of cell therapy in the future.

## 7. Conclusions

Cell therapy offers a promising regenerative medicine approach for treating ischemic diseases (such as stroke, myocardial infarction, and peripheral artery disease), aiming to overcome the limitations of traditional drug and surgical therapies. This review systematically explored the application potential of various stem cell types (including NSCs, ESCs, iPSCs, MSCs) and their complex mechanisms of action, covering aspects such as direct differentiation, paracrine signaling, immune modulation, and microenvironment remodeling. Although clinical trials have initially demonstrated the potential of stem cells to improve tissue repair and function, they have also revealed challenges related to cell survival, delivery efficiency, standardization, and therapeutic consistency. Gene editing technologies and bioengineering innovations (such as advanced biomaterials, engineered exosomes) are being actively explored to enhance the therapeutic properties, targeting, and persistence of cells. However, safety concerns (such as off-target risks, tumor formation), immunogenicity, large-scale production, and ethical considerations remain critical obstacles that must be addressed before widespread clinical application can be achieved. In the future, continued basic research, technological optimization, and rigorously designed large-scale clinical trials are crucial for overcoming existing challenges, validating long-term efficacy and safety, and ultimately promoting the clinical translation of precise, personalized cell therapy strategies in the field of ischemic diseases.

## Figures and Tables

**Figure 1 ijms-26-06320-f001:**
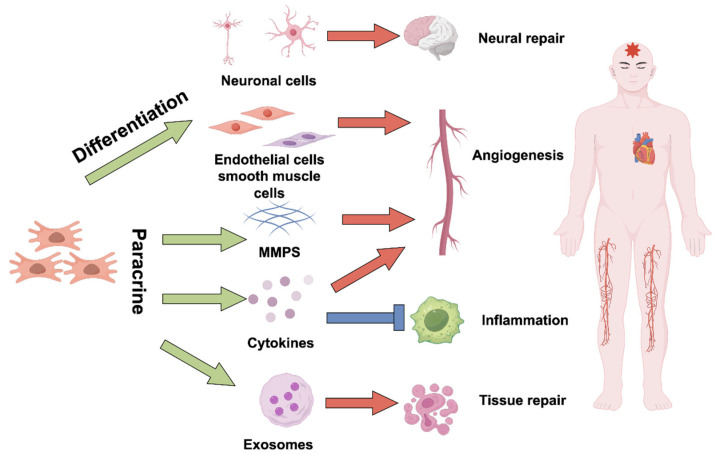
The mechanisms of stem cell therapy for ischemia. Stem cells promote the repair and regeneration of damaged tissue in ischemic diseases through multiple mechanisms. They can directly differentiate into neurons, endothelial cells, and smooth muscle cells, participating in neural repair and angiogenesis. Additionally, through paracrine effects, stem cells release various factors—such as matrix metalloproteinases, cytokines, and exosomes—that regulate the tissue microenvironment, promote vascular formation, reduce inflammation, modulate immunity, inhibit apoptosis, suppress oxidative stress, and remodel the extracellular matrix. These mechanisms work together to create favorable conditions for the repair and functional recovery of ischemic tissues.

**Figure 2 ijms-26-06320-f002:**
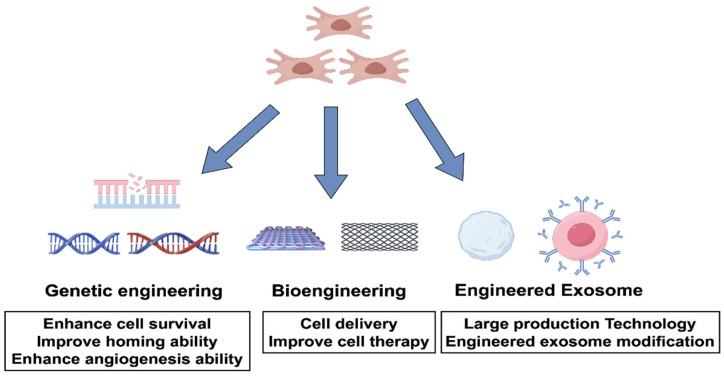
Future directions of stem cell therapy. To further improve the efficacy of stem cell therapy for ischemic diseases, future research will primarily focus on optimizing stem cells and their products using advanced techniques such as genetic engineering, bioengineering, and exosome engineering. Genetic engineering can enhance cell survival, homing ability, and pro-angiogenic capacity. Bioengineering can improve cell delivery methods, thereby increasing the precision and effectiveness of cell therapy. Exosome engineering will not only advance large-scale exosome production techniques but also expand the potential of exosome applications in disease treatment through engineering modifications. These innovative strategies offer broad prospects for the clinical translation and efficacy improvement of stem cell therapy.

**Table 1 ijms-26-06320-t001:** Drugs for treating ischemia.

Drug Category	Mechanism of Action	Drug Names	Disadvantages
Anticoagulants	Inhibit platelet aggregation and limit thrombus formation	Heparin, Warfarin, Rivaroxaban, Apixaban	Risk of bleeding
Thrombolytics	Dissolve already formed thrombi	Alteplase, Retaplase	Must be used within 4.5 h of onset; bleeding risk
Vasodilators	Dilate blood vessels to increase blood supply	Nitroglycerin	Hypotension, dizziness, headache
Calcium Channel Blockers	Block calcium ion channels, dilate vessels	Nifedipine, Diltiazem	Bradycardia, hypotension, not suitable for some heart failure patients
Neuroprotective Agents	Protect neurons from ischemia-reperfusion injury	Edaravone	Must be used within 48 h of onset; limited clinical efficacy
Antiplatelet drugs	Inhibit platelet activation and aggregation	Aspirin, Clopidogrel, Tirofiban	Risk of bleeding

**Table 2 ijms-26-06320-t002:** Comparison of Different Types of Stem Cells.

Stem Cell Type	Advantages	Disadvantages	Sources	Challenges
NSCs	Can differentiate into various neural cell types; potential for brain injury repair	Limited availability and ethical concerns	Brain tissue	Integration and survival in host tissue
ESCs	Pluripotent, can differentiate into any cell type	Ethical concerns, risk of teratoma formation	Embryos	Ethical issues, tumor risk
IPSCs	Pluripotent; derived from adult cells, avoiding ethical concerns;	Genetic instability, potential for tumor formation	Adult somatic cells	Controlled differentiation, safety
MSCs	Easy to obtain; immunomodulatory properties; lower ethical concerns	Limited multilineage potential; variable quality	Bone marrow, adipose tissue	Standardization, variability in patient sources
HESs	Can differentiate into all types of mature blood cells and EPC	Restricted availability, complex procurement and processing	Bone marrow, umbilical cord blood	limited cell sources
CSCs	Can differentiate into cardiomyocytes, facilitating myocardial regeneration	low abundance of true CSCs in heart tissue	heart tissue	Difficulty in accurate identification and purification

**Table 3 ijms-26-06320-t003:** Clinical trials and outcomes of stem cell therapy for ischemic diseases.

Ischemic Disease	Cell Types Used	Main Clinical Outcomes	Delivery Routes
Acute Myocardial Infarction	BM-MSCs, UC-MSCs, HSCs, CSCs	Improved cardiac function: ~4.58% increase in LVEF, ~5.18 mL reduction in LVESV; dose-dependent effects	Intracoronary injection, intramyocardial injection, intravenous injection
Ischemic Stroke	BM-MSCs	Some functional neurological improvements reported; mixed results across trials	Intravenous injection, arterial injection, intracerebral injection
Chronic Limb Ischemia	B-MSCs, AD-MSCs Placental MSCs	Improved blood perfusion, ulcer healing, pain relief, reduced amputation rates	Arterial injection, intravenous injection, intramuscular injection

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
