# Peer review of "Stem Cell Therapy Approaches for Ischemia: Assessing Current Innovations and Future Directions"

_ijms, 2025, doi:10.3390/ijms26136320_

Round 1
Reviewer 1 Report
Comments and Suggestions for Authors
In the present review, Changguo Ma and collegues aim to provide a comprehensive overview of the current state of stem cell therapy in ischemic diseases. It examines the therapeutic potential of various stem cell types, including in their report mesenchymal stem cells (MSCs), induced pluripotent stem cells (iPSCs), embryonic stem cells (ESCs), and neural stem cells (NSCs) paying particular attention to the molecular and cellular mechanisms by which these cells contribute to tissue repair. The authors also discussed recent innovations in gene editing, biomaterials, and cell delivery systems that enhance therapeutic efficacy. Additionally, the review discusses current clinical trial data, challenges in clinical translation, and future directions in the field, describing the development of a safe, effective, and personalized regenerative therapies for ischemic conditions.
Overall, the manuscript represents a good starting point to understand the impact of stem cells in mechanism of tissue repair/regeneration after damage. I have only a main consideration to ameliorate the impact of the manuscript itself. I suggest to the authors to focus their attention on all cell types described till now, in order to give a picture of the endogenous regenerative potential of the adult heart. I would recommend including a dedicated paragraph discussing the role of hematopoietic stem cells and cardiac stem cells in the treatment of ischemic diseases, since the latter cells are resident within the heart and widely used as source for cell therapies and treatment of cardiac diseases. It is essential to maintain a broad perspective and an open mind in the field of stem cell therapy, ensuring that all types of stem cells—particularly cardiac stem cells—are considered and discussed in the context of cardiac repair. While other regenerative approaches are well described, the specific potential of cardiac stem cells and hematopoietic stem cells —including their mechanisms of action, current clinical trial status, and limitations—is not addressed.
(see doi: 10.1242/dev.124.10.1929; https://doi.org/10.1007/978-1-61779-815-3_19; https://doi.org/10.1038/cdd.2017.130; DOI: 10.1007/978-3-030-24108-7_8; doi: 10.1038/s42003-021-02677-y ;). It could be helpful to include them in the cartoon.
Moreover, the authors should better discuss the role of gene editing in enhancing therapeutic efficacy of stem cells in cardiovascular diseases (please see : doi: 10.1080/14712598.2018.1430762; https://doi.org/10.1161/CIR.0000000000001296; https://doi.org/10.1111/joim.13308)
Author Response
Thank you very much for taking the time to review this manuscript. Please find the detailed responses below and the corresponding revisions highlighted in track changes in the resubmitted files.
Comments 1: Overall, the manuscript represents a good starting point to understand the impact of stem cells in mechanism of tissue repair/regeneration after damage. I have only a main consideration to ameliorate the impact of the manuscript itself. I suggest to the authors to focus their attention on all cell types described till now, in order to give a picture of the endogenous regenerative potential of the adult heart. I would recommend including a dedicated paragraph discussing the role of hematopoietic stem cells and cardiac stem cells in the treatment of ischemic diseases, since the latter cells are resident within the heart and widely used as source for cell therapies and treatment of cardiac diseases. It is essential to maintain a broad perspective and an open mind in the field of stem cell therapy, ensuring that all types of stem cells—particularly cardiac stem cells—are considered and discussed in the context of cardiac repair. While other regenerative approaches are well described, the specific potential of cardiac stem cells and hematopoietic stem cells —including their mechanisms of action, current clinical trial status, and limitations—is not addressed.
Response 1: Thank you for bringing this to our attention. We fully agree with this comment. Therefore, we have added a paragraph describing hematopoietic stem cells and cardiac stem cell therapy for ischemia in the revised manuscript. The paragraph is located in Section 4, page 7, lines 259-304, and is highlighted in red. In Clinical Results 5.1, we expanded clinical studies on the treatment of myocardial ischemia with HCSs and CSCs, specifically the sentences in section 5.1, page 11, lines 408-417, highlighted in red.
Comments 2: Moreover, the authors should better discuss the role of gene editing in enhancing therapeutic efficacy of stem cells in cardiovascular diseases.
Response 2: Thank you for pointing this out. We fully agree with this comment. We have added content regarding gene editing promoting CSC therapy for myocardial ischemia, specifically gene editing promoting CSC survival and promoting CSC paracrine secretion. This is located in section 6.1, page 14, lines 529-540 and page 15, lines 559-564.
Reviewer 2 Report
Comments and Suggestions for Authors
In the manuscript entitled “Stem Cell Therapy Approaches for Ischemia: Assessing Current Innovations and Future Directions” the authors explore the potential of cell therapy for ischemic diseases, moving beyond the limitations of traditional treatments. From direct differentiation to paracrine effects, stem cells offer new avenues for tissue repair. Despite challenges like cell survival and standardization, advancements in genetic engineering and bioengineering promise to unlock the full potential of this regenerative medicine. The review is well-written and provides a comprehensive overview of cell therapy for ischemic diseases, highlighting its potential, mechanisms of action, and future directions.
The manuscript could be improved with some issues to take into account.
Since the general introduction of the article already covers ischemic diseases, the current title for Section 2 might be slightly redundant. Consider alternative titles.
You state: "...the drugs used clinically can be broadly classified into the following categories: thrombolytics, antiplatelet drugs, anticoagulants…" However, there isn't a specific subsection dedicated to "Antiplatelet drugs." This inconsistency should be resolved either by adding the subsection or adjusting the introductory sentence.
The presence of the progesterone receptor (PR) in iPSCs is crucial because, given its importance in the differentiation of embryonic stem cells, it's presumed to modulate the differentiation processes of iPSCs towards specific cell types. This directly influences their capacity to generate cells useful for tissue repair, suggesting PR as a key factor determining the properties and therapeutic efficacy of iPSCs in regenerative medicine. A specific focus on this, 10.1007/s12015-024-10776-6, would further enhance the manuscript.
Production costs and scalability are briefly mentioned as challenges, but the review doesn't elaborate on the immense economic and accessibility implications that cell therapy will entail. Expanding on these aspects would offer a more comprehensive perspective.
Minor English editing and grammatical check.
Author Response
Thank you very much for taking the time to review this manuscript. Please find the detailed responses below and the corresponding revisions/corrections highlighted/in track changes in the resubmitted files.
Comments 1: Since the general introduction of the article already covers ischemic diseases, the current title for Section 2 might be slightly redundant. Consider alternative titles.
Response 1: Thank you for bringing this to our attention. We fully agree with this comment. Therefore, we have replaced the title with "Overview and Pathogenesis of Ischemic Diseases," located on page 2, line 53, highlighted in red.
Comments 2: You state: "...the drugs used clinically can be broadly classified into the following categories: thrombolytics, antiplatelet drugs, anticoagulants…" However, there isn't a specific subsection dedicated to "Antiplatelet drugs." This inconsistency should be resolved either by adding the subsection or adjusting the introductory sentence.
Response 2: Thank you for pointing this out. We fully agree with this comment. We have added a description of antiplatelet drugs in section 3.5, page 4, lines 163-177.
Comments 3: The presence of the progesterone receptor (PR) in iPSCs is crucial because, given its importance in the differentiation of embryonic stem cells, it's presumed to modulate the differentiation processes of iPSCs towards specific cell types. This directly influences their capacity to generate cells useful for tissue repair, suggesting PR as a key factor determining the properties and therapeutic efficacy of iPSCs in regenerative medicine. A specific focus on this, 10.1007/s12015-024-10776-6, would further enhance the manuscript.
Response 3: Thank you for your insightful feedback. We are in full agreement that the progesterone receptor (PR) plays a crucial role in iPSCs. Given its pivotal function in the differentiation of embryonic stem cells, it is highly likely that PR also modulates the differentiation of iPSCs. This modulation directly impacts their ability to generate cells that are valuable for tissue repair. Therefore, PR is a key determinant of the properties and therapeutic efficacy of iPSCs in regenerative medicine. To address this important aspect, we have expanded the discussion on PR and its role in iPSC differentiation for treating ischemia in section 4.1, on page 6, lines 248-254, highlighted in red.
Comments 4: Production costs and scalability are briefly mentioned as challenges, but the review doesn't elaborate on the immense economic and accessibility implications that cell therapy will entail. Expanding on these aspects would offer a more comprehensive perspective.
Response 4: Thank you for pointing this out. We fully agree with this comment. We have added a paragraph describing the production costs and accessibility of stem cells, highlighted in red in Section 6.5, page 17, lines 647-688.
Comments 5: Minor English editing and grammatical check.
Response 5: Thank you for pointing this out. We fully agree with this comment. We have reviewed the entire text for grammar and corrected minor errors.
Round 2
Reviewer 1 Report
Comments and Suggestions for Authors
he authors have addressed all my concerns. The review is now ready for publication